# *In silico* investigation of novel *Plasmodium Falciparum* glycogen synthase kinase (*pf*GSk3β) inhibitors for the treatment of malaria infection

Kassim Folorunsho Adebambo ⓘ*, Sara Otify ⓘ

Department of Clinical, Pharmaceutical and Biological Sciences, University of Hertfordshire, Hatfield, United Kingdom

* k.adebambo@herts.ac.uk

## Abstract

Malaria, a parasitic disease, remains a major global health concern, with over 260 million cases reported worldwide in 2023. As resistance to current antimalarial drugs increases, the demand for ongoing research into new therapeutic targets and strategies grows. Glycogen synthase kinase (pfGSK3β) is a crucial enzyme involved in metabolic processes of the malaria parasite. In this research, an in silico study was conducted to explore this enzyme as a potential target for drug repurposing. A Python program was used to mine and extract data from the CHEMBL database, which yielded 53 potential GSK-3β inhibitors. Subsequent in silico studies included molecular docking, molecular dynamics simulations (MD, run at 100 ns on GROMACS 2023 1), and molecular mechanics Poisson-Boltzmann surface area (MMPBSA). In silico data analysis identified three potential drug molecules: S20-CHEMBLID 1910196 (4-[5-(6- hydroxy- 1H-indol-2- yl)pyridin-3- yl]benzonitrile), S39-CHEMBL ID 2321945 (2-(7- bromo- 2- hydroxy- 1H-indol-3-yl)-3- oxoindole- 6- carboxylic acid), and S56-CHEMBL ID 2321951 (methyl 2-(2- hydroxy-1H-indol-3-yl)-3-nitroso-1H-indole-5-carboxylate),which could inhibit pfgsk 3β. Compound S56 demonstrated better in silico performance than S1 – (3,6- diamino- 4-(2- chlorophenyl)thieno[2, 3- b]pyridine- 2, 5-,5-dicarbonitrile), the co-crystallised ligand in pfgsk 3 β used as a control. The binding affinities of S1 and S56 are- 7.1157074 (10 ligand interactions) and − 5.64057302 (12 ligand interactions), respectively. The MD runs yielded average root-mean-square deviations (RMSDs) of 4.5 nm for S1 and 1.0 nm for S56. Furthermore, the root mean square fluctuation (RMSF) of S1 showed greater fluctuation between 0–1000 atoms compared to S56. MMPBSA analysis revealed comparable total energies: S56 was −14.45 kj/mol and S1 was −13.03 kj/mol. An in silico toxicity study using Protox III indicated the possible toxicity of the repurposed compounds. In conclusion, we propose that molecules S39, S20, and S56 could be repurposed as potential anti-malaria drugs.

**Data availability statement:** All relevant data are within the paper and its Supporting Information files.

**Funding:** The author(s) received no specific funding for this work.

**Competing interests:** The authors have declared that no competing interests exist.

## Introduction

Malaria is a disease caused by a parasitic protozoan called Plasmodium. There are four common species of Plasmodium: P. falciparum, P. vivax, P. ovale, and P. malariae. The most prevalent is Plasmodium falciparum, which invades a large number of red blood cells, making malaria a severe and potentially life-threatening disease [1]. Hassan et al.[2] suggested that malaria could be considered an inflammatory disease because the malaria parasite can trigger an inflammatory response similar to that seen in other pathogenic diseases. The malaria parasite is transmitted to humans by infected female mosquitoes (Anopheles). It is found across approximately 83 countries, mainly in tropical and subtropical regions, including Africa, Asia, Central and South America, parts of the Middle East, and some specific islands. In 2023, there were over 260 million cases of malaria and more than 590,000 deaths worldwide, with 76% of these affecting children under 5 years of age [3]. Temperatures between 20 and 30 degrees Celsius, high humidity, seasonal rainfall, forest ecosystems, and rural areas all increase the risk of exposure to malaria. Although the highest risk of mosquito bites occurs at dawn and dusk, the danger exists at any time of day, whether indoors or outdoors [4]. Lacerda-Queiroz and co-worker [5] reported that a single malaria infection can lead to high morbidity and death if not properly managed.

### Treatment/resistance

There are key factors to consider when beginning treatment. For example, it is crucial to identify the Plasmodium species because different species can cause varying severities of malaria. For instance, P. falciparum can rapidly lead to a life-threatening condition, while other species may be less severe. Additionally, P. vivax and P. Ovale require treatment targeting the dormant hypnozoites to prevent relapse later. Based on the symptoms, malaria is classified as either uncomplicated or severe, which guides treatment choices. Assessing the likelihood of drug resistance and its patterns depends on knowledge of the geographical area, aiding in selecting the most appropriate treatment or combination of treatments. However, the malaria parasite has been found to develop resistance rapidly to commonly used drugs in clinics. For example, malaria has shown swift resistance to chloroquine and other drugs like artemisinin [6]. Shockingly, Nicholas J White's report [7] stated that, although malaria infection had developed resistance to most frontline drugs, hope remained in Artemisinin, and he warned that losing Artemisinin to resistance could render malaria untreatable. Li and coworkers [6] have demonstrated that resistance to Artemisinin is now emerging. Consequently, there is an urgent need to investigate new targets and develop novel drug molecules for the treatment of malaria.

### Glycogen synthase kinase and its role in malaria

Glycogen synthase kinase-3 (GSK-3β) is a serine/threonine protein kinase with two isoforms. It was found to phosphorylate glycogen synthase but also plays a role in many biological processes [8], including cell proliferation, differentiation, and protein synthesis [9]. Serine is a non-essential amino acid involved in cell growth and the

synthesis of purine, adenine, and guanine bases in DNA [10]. Threonine is an amino acid that supports fat metabolism in the liver [11]. It has been shown that GSK3β, along with other protein kinases, is essential for the parasite's ability to proliferate within red blood cells. However, in P. falciparum, pfGSK-3β is vital for schizogony, and within infected red blood cells, it undergoes phosphorylation at tyrosine 229. Because it is hypothesised that GSK-3β in P. falciparum plays roles in cell cycle control, differentiation, and metabolic regulation, it is considered a potential drug target to combat resistance to antimalarial drugs [9]. The genome of the malaria parasite P. falciparum encodes approximately 65 eukaryotic protein kinases [12]; some of these kinases contribute to the parasite's process of infecting host cells [13]. To date, only one anti-malarial drug has been successfully developed to target a kinase enzyme. This drug targets the PI4K enzyme and is well tolerated, with very high antimalarial efficacy. Currently, the PI4K inhibitor is in clinical trials [14,15]. Additionally, the other enzyme studied is pfGSK3β, with current research focusing on the investigation of novel inhibitors for pfGSK3β.

In the malaria life cycle, before RBC invasion, it has been reported that PfGSK3β phosphorylates antigen one, a microneme-localised secreted protein that mediates the essential step of "tight junction" formation with the host cell membrane [16]. Recent studies provide evidence that protein kinases play a crucial role in gametocytogenesis and the subsequent formation of gametes [13]. This development takes approximately 10 days and is accompanied not only by distinct morphological changes but also by extensive reprogramming of the parasite's cellular metabolism [17]. The importance of kinases in this process offers additional potential drug targets for developing transmission-blocking chemotypes [17]

## Target selection

GSK-3β in Plasmodium falciparum (pfGSK3β) has been shown to support parasite survival by altering red blood cell metabolism, membrane transport, and cytoskeletal properties, potentially enhancing parasite growth. The protein kinase may also promote the upregulation of conductive and new permeation pathways. Furthermore, it has been demonstrated that interference with circadian clock regulators can facilitate the maturation and replication of the blood stages of parasites [9]; therefore, targeting this protein could be an effective strategy against malaria. The 3D structure of pfGSK-3β is illustrated below in Fig 1, and the various molecules synthesised and tested by Masch and co-workers [9] as pfGSK3β inhibitors are shown in Fig 2.

## Challenges in malaria management

As previously noted, malaria is among the most life-threatening diseases, posing a significant health challenge. Several barriers impede effective management of the disease, including resistance to anti-malarial drugs such as chloroquine,

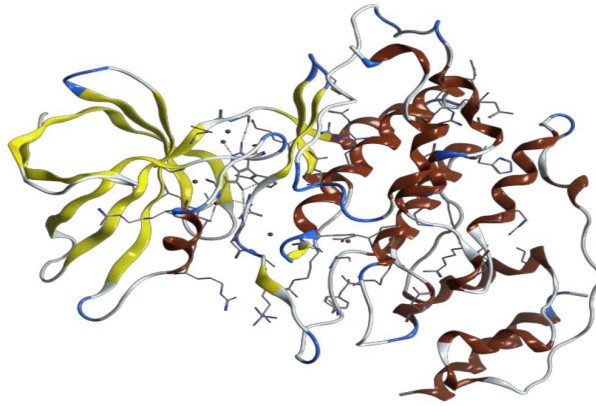

**Fig 1. 3D structure of plasmodium falciparum glycogen synthase-3, PDB ID: 3ZDI [18].**

**Fig 2. Possible drug molecules that have been investigated for the inhibition of pfGSK3β by Masch and co-workers [9].**

the use of similar chemical derivatives leading to cross-resistance, and genetic mutations in parasites. [19]. These issues emphasise the need for ongoing research to discover new drugs and therapeutic targets for the effective treatment of malaria infection.

## Aim of the research

This research was inspired by the work of Masch and co-workers [9], who began developing a new drug molecule targeting a novel site in the malaria parasite, pfgsk3β. Most of the compounds synthesised and tested by the group, are

shown in Fig 2, and these molecules serve as a template for our in silico investigation. The use of computational studies is increasingly important in the era of machine learning to accelerate the development of new drug molecules. Therefore, using the potential drug molecules and targets reported (Fig 2) as a reference, we explored the possibility of repurposing existing Gsk3β inhibitors from the CHEMBL database as pfgsk3β inhibitors through computational and data science tools like Python. This in silico approach is crucial, given the high risks, costs, environmental impacts, and time required to develop a new drug from scratch. Our approach aligns with the findings of Wu and co-workers [20]. and Parvantaneni [21], who reported various successes achieved through in silico research. Moreover, computational investigations have played a significant role in repurposing existing biomolecules for new disease treatments. For example, the use of a scoring and ranking model [22], and structure-based drug repurposing research using in silico tools has been reported by Choudhury and co-worker [23].,

Finally, we aimed to use Python to generate code for searching the ChEMBL database for existing GSK-3 inhibitors, investigate the binding affinity of bioactive molecules identified through the Python search in pfGSK3β using PDB 3ZDI, and validate this drug repositioning study. The binding affinity will be compared with that of the molecules in Fig 2. Furthermore, molecular dynamics simulations of the potential drug compounds will be carried out using GROMACS tools, along with in silico toxicity prediction via the web server Protox III. The data from these computational studies will help determine whether a new potential drug molecule has been identified for inhibiting pfGSK3β.

## Methodology

**Data mining study on CHEMBL database using python coding.** Python is a programming language used for data science and machine learning [24]. The Python tool has become highly valuable for drug repurposing, enabling scientists to extract insights from extensive databases for drug repositioning studies. The Python used in this research was installed via the Anaconda platform. Anaconda Navigator was set up using this weblink: https://www.anaconda.com/download/success.

The ChEMBL database is an open resource that contains many drugs, such as bioactive compounds, along with their biological activities, functional data, and ADMET information [25]. According to Michal Nwotka [26], Python software has been developed to facilitate data mining within the CHEMBL database. This Python tool is user-friendly because the CHEMBL database offers a client library API for straightforward data extraction. A typical data mining procedure, adapted by our team, is provided in Appendix 1 in S1 File.

Beginning with the coding steps outlined in Appendix 1 in S1 File, we compiled codes to identify all GSK3β inhibitors in the CHEMBL database. The flowchart for the coding process we created is presented in Fig 3. The data mining process resulted in 53 potential inhibitors.

## Molecular docking

Our research was inspired by the work reported by Masch and co-worker [9]. In their report, they showed that some molecules inhibited pfGSK3β, a novel target for treating malaria. We used the designed molecules [Fig 2] as a control for our in silico research. Therefore, 65 compounds were used in the molecular docking experiment, twelve from the Masch et al., 2015 compounds, and 53 from data mining of the CHEMBL library. The molecular docking was performed using Molecular Operating Environment (MOE) software, a comprehensive system utilised by medicinal chemists, biologists, crystallographers, and computational chemists [27]. The general flowchart for molecular docking, as reported by Morris and co-workers [28], is shown in Fig 4 below.

## Ligand preparation

As shown in Fig 5, ligands need to be prepared before conducting the molecular docking study. Ligand preparation is a crucial step in protein-ligand interactions, as it involves synthesising ligands and minimising their energies, thereby making

## Data Mining of Chembl Library using Python Code

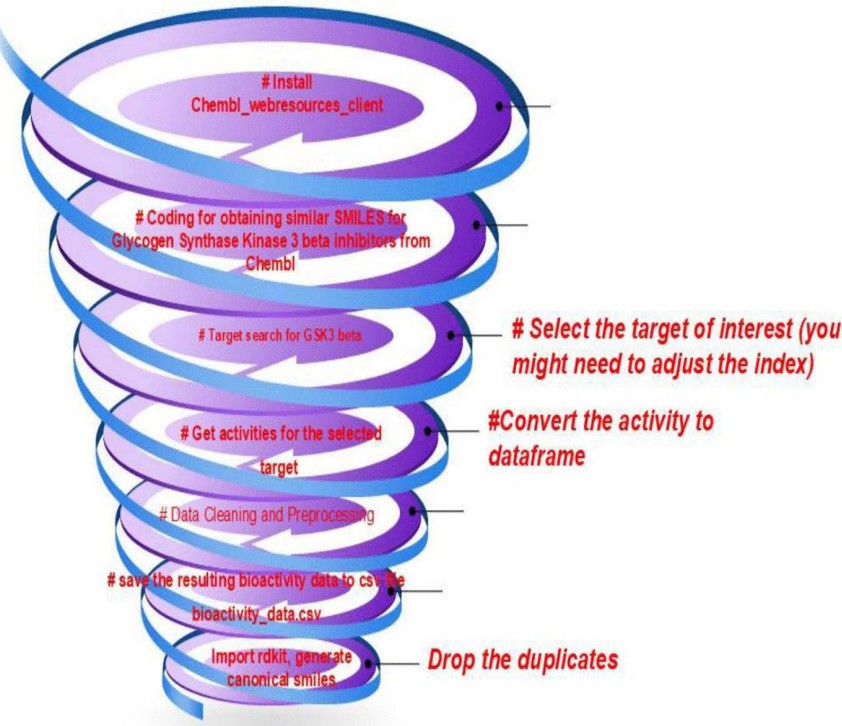

# Install Chembl_webresources_client

# Coding for obtaining similar SMILES for Glycogen Synthase Kinase 3 beta inhibitors from Chembl

# Target search for GSK3 beta

**# Select the target of interest (you might need to adjust the index)**

# Get activities for the selected target

**#Convert the activity to dataframe**

# Data Cleaning and Preprocessing

# save the resulting bioactivity data to csv file bioactivity_data.csv

Import rdkit, generate canonical smiles

**Drop the duplicates**

**Fig 3. Steps followed during Data Mining protocol of the Chembl Database using Python tool.**

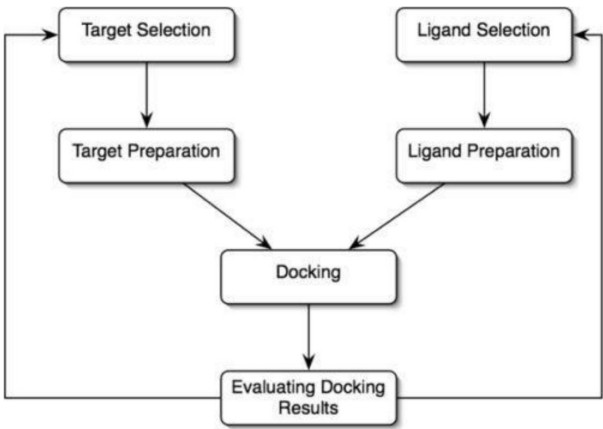

**Fig 4. Steps involved in docking [28].**

them suitable for effective protein-ligand interactions. During preparation, SMILES of the molecules were input into MOE using the builder command, and the final structures were then minimised. The prepared ligand structures are converted into an.mdb file (a format suitable for MOE docking software).

## Protein preparation

A 3D model of pfGSK3β (pdb ID 3ZDI) was downloaded from the Protein Data Bank website [18]. The protein was prepared for docking using the quick prep ion in MOE software. The preparation involved preserving the sequence, neutralising the receptor, and removing all water molecules beyond 4.5Å from the ligand or receptor. Additionally, all missing amino acids were fixed. The protein was refined to an RMS gradient of 0.1 kcal/mol/å². The site finder tool in MOE was used to identify the binding pocket of the co-crystallised ligand, which served as the target site for the binding of the repurposed drug molecules.

## Molecular docking- protein-ligand interaction

All 65 prepared molecules were docked into the identified binding pocket of the co-crystallised ligand. For this experiment, a triangular placement matcher was used, and docking was performed with rigid receptor refinement. Additionally, 30 binding poses were generated (London dG-30 poses), and the system was set to return the five best poses (GBVI/WSA dG-5 poses). Rigid receptor placement means that the receptor or protein remains fixed while the ligand is flexible to undergo conformational changes, but the protein's structure does not adjust.

## Molecular dynamics simulations study

The Molecular Dynamics simulations were conducted using GROMACS 2023.1 on a GPU, following the step-by-step procedure outlined by Lemkul and co-worker [29] for the Protein-Ligand Complex. The protein 3ZDI obtained from the PDB had an inbuilt amino acid code that required resolution because it was not included in the forcefield residue data. This issue was resolved using the Protein Repair and Analysis Server (Protein Repair & Analysis Server). The server-generated PDB output was subsequently used to re-dock the 65 ligands to assess whether the repaired PDB structure would influence docking scores and ligand ranking. Rescoring revealed no changes in the types or numbers of ligand interactions among the 65 molecules. This confirmed that the repair did not impact our docking results. The molecular dynamics simulations employed the CHARMM all-atom force field to generate the protein topology and used the official Cgenff server for ligand topology. The production run of the molecular dynamics simulations was carried out over 100 ns.

## g_mmpbsa calculation

The binding energy calculations for the protein-ligand complexes were further analysed using the Molecular Mechanics Poisson-Boltzmann Surface Area (MM-PBSA) method, which was performed with the g_mmpbsa tool [30]. The g_mmpbsa analysis was conducted over a 100 ns duration of the molecular dynamics simulations.

## In silico toxicity study

Drug development involves evaluating potential analogues for safety, with toxicity levels being a crucial criterion. Moving a potential drug molecule from the laboratory to the market requires substantial investment, and many drug compounds never progress beyond toxicity testing [31]. Most drugs fail pre-clinical tests both in vivo and in vitro; therefore, employing in silico pre-clinical trials to screen drugs for potential toxicity is a valuable approach before committing to in vivo studies. This method saves time and reduces the environmental stress associated with wet-laboratory experiments. The use of Protox III has been developed to support clinical research. Consequently, our in silico toxicity study was carried out using a web server service, ProTox-3.0 – Prediction of TOXicity of chemicals [32]. The online tool was utilised to perform in silico analysis of the potential drug candidates investigated in this research.

## Results and discussion

This research aimed to use Python to mine the CHEMBL library for potential GSK3β inhibitors that could be repurposed to inhibit pfgsk3β. The code was successfully developed; this data-mining process generated 53 molecules ranked by IC50 (Table 1).

**Table 1. 53 molecules generated from the data mining of the CHEMBL database using the generated coding.**

| Chembl_id | Canonical_smiles | Standard value (1C50nM) | Structure No. |
|---|---|---|---|
| CHEMBL479170 | Oc1ccc(-c2cncc(-c3cc4cc(O)ccc4[nH]3)c2)cc1 | 40000 | **13** |
| CHEMBL1910190 | N#Cc1ccc(-c2cncc(-c3cc4cc(O)ccc4[nH]3)c2)cc1 | 100000 | **14** |
| CHEMBL1910191 | CS(=O)(=O)c1ccc(-c2cncc(-c3cc4cc(O)ccc4[nH]3)c2)cc1 | 7300 | **15** |
| CHEMBL1910192 | Oc1ccc(-c2cncc(-c3cc4cc(O)ccc4[nH]3)n2)cc1 | 4700 | **16** |
| CHEMBL1910193 | N#Cc1ccc(-c2cncc(-c3cc4cc(O)ccc4[nH]3)n2)cc1 | 26000 | **17** |
| CHEMBL1910194 | CS(=O)(=O)c1ccc(-c2cncc(-c3cc4cc(O)ccc4[nH]3)n2)cc1 | 26000 | **18** |
| CHEMBL1910195 | Oc1ccc(-c2cncc(-c3cc4ccc(O)cc4[nH]3)c2)cc1 | 1100 | **19** |
| CHEMBL1910196 | N#Cc1ccc(-c2cncc(-c3cc4ccc(O)cc4[nH]3)c2)cc1 | 4000 | **20** |
| CHEMBL1910197 | CS(=O)(=O)c1ccc(-c2cncc(-c3cc4ccc(O)cc4[nH]3)c2)cc1 | 1100 | **21** |
| CHEMBL1910198 | Oc1ccc(-c2cncc(-c3cc4ccc(O)cc4[nH]3)n2)cc1 | 5300 | **22** |
| CHEMBL1910199 | N#Cc1ccc(-c2cncc(-c3cc4ccc(O)cc4[nH]3)n2)cc1 | 5700 | **23** |
| CHEMBL1910200 | CS(=O)(=O)c1ccc(-c2cncc(-c3cc4ccc(O)cc4[nH]3)n2)cc1 | 2300 | **24** |
| CHEMBL2321963 | O=C1Nc2c(cccc2C(F)(F)F)/C1=C1/Nc2cc(C(=O)O)ccc2/C1=N\O | 10000 | **25** |
| CHEMBL2321962 | O=C1Nc2c(Br)cccc2/C1=C1/Nc2cc(C(=O)O)ccc2/C1=N\O | 10000 | **26** |
| CHEMBL2321960 | COC(=O)c1ccc2c(c1)NC(=C1\C(=O)Nc3c(Br)cccc31)/C2=N/O | 10000 | **27** |
| CHEMBL2321958 | CN1C(=O)/C(=C2\Nc3ccc(C(=O)O)cc3\C2=N/O)c2cccc(Br)c21 | 10000 | **28** |
| CHEMBL2321957 | COC(=O)c1ccc2c(c1)C(=N\O)/C(=C1/C(=O)N(C)c3c(Br)cccc31)N2 | 10000 | **29** |
| CHEMBL1233659 | O=C1Nc2c(Br)cccc2/C1=C1/Nc2ccc(C(=O)O)cc2/C1=N\O | 10000 | **30** |
| CHEMBL2321956 | O=C1Nc2c(cccc2C(F)(F)F)/C1=C1/Nc2ccc(-c3nnn[nH]3)cc2/C1=N\O | 10000 | **31** |
| CHEMBL2321955 | N#Cc1ccc2c(c1)C(=N\O)/C(=C1/C(=O)Nc3c1cccc3C(F)(F)F)N2 | 10000 | **32** |
| CHEMBL2321954 | COC(=O)c1ccc2c(c1)C(=N\O)/C(=C1/C(=O)Nc3c1cccc3C(F)(F)F)N2 | 10000 | **33** |
| CHEMBL2321953 | COC(=O)c1ccc2c(c1)C(=N\O)/C(=C1/C(=O)Nc3c(Br)cccc31)N2 | 10000 | **34** |
| CHEMBL213454 | Cn1c(O)c(-c2[nH]c3cccccc3c2N=O)c2cccc(Br)c21 | 10000 | **35** |
| CHEMBL2321949 | O=C1Nc2c(cccc2C(F)(F)F)/C1=C1/Nc2ccccc2/C1=N\O | 10000 | **36** |
| CHEMBL2321947 | O=C1Nc2c(Br)cccc2/C1=C1/Nc2ccc(CO)cc2C1=O | 10000 | **37** |
| CHEMBL2321946 | O=C1Nc2c(cccc2C(F)(F)F)/C1=C1/Nc2cc(C(=O)O)ccc2C1=O | 10000 | **38** |
| CHEMBL2321945 | O=C1Nc2c(Br)cccc2/C1=C1/Nc2cc(C(=O)O)ccc2C1=O | 10000 | **39** |
| CHEMBL2322019 | COC(=O)c1ccc2c(c1)N/C(=C1\C(=O)Nc3c1cccc3C(F)(F)F)C2=O | 10000 | **40** |
| CHEMBL2321975 | COC(=O)c1ccc2c(c1)N/C(=C1\C(=O)Nc3c(Br)cccc31)C2=O | 10000 | **41** |
| CHEMBL2321974 | O=C1Nc2ccccc2/C1=C1/Nc2cc(C(=O)O)ccc2C1=O | 10000 | **42** |
| CHEMBL2321973 | COC(=O)c1ccc2c(c1)C(=O)/C(=C1/C(=O)N(C)c3c(Br)cccc31)N2 | 10000 | **43** |
| CHEMBL2321972 | O=C1Nc2c(cccc2C(F)(F)F)/C1=C1/Nc2ccc(-c3nnn[nH]3)cc2C1=O | 10000 | **44** |
| CHEMBL2321971 | N#Cc1ccc2c(c1)C(=O)/C(=C1/C(=O)Nc3c1cccc3C(F)(F)F)N2 | 10000 | **45** |
| CHEMBL2321970 | COC(=O)c1ccc2c(c1)C(=O)/C(=C1/C(=O)Nc3c1cccc3C(F)(F)F)N2 | 10000 | **46** |
| CHEMBL2321967 | COC(=O)c1ccc2c(c1)C(=O)/C(=C1/C(=O)Nc3ccccc31)N2 | 10000 | **47** |
| CHEMBL375870 | Cn1c(O)c(C2=Nc3ccccc3C2=O)c2cccc(Br)c21 | 10000 | **48** |
| CHEMBL373834 | O=Nc1c(-c2c(O)[nH]c3c(Br)cccc23)[nH]c2ccccc12 | 10000 | **49** |
| CHEMBL2321969 | COC(=O)c1ccc2c(c1)C(=O)/C(=C1/C(=O)Nc3c(Br)cccc31)N2 | 25000 | **50** |
| CHEMBL2321959 | O=C1Nc2ccccc2/C1=C1/Nc2cc(C(=O)O)ccc2/C1=N\O | 6100 | **51** |
| CHEMBL2321878 | O=C1Nc2c(cccc2C(F)(F)F)/C1=C1/Nc2ccc(C(=O)O)cc2/C1=N\O | 1200 | **52** |
| CHEMBL2321968 | O=C1Nc2ccccc2/C1=C1/Nc2ccc(C(=O)O)cc2C1=O | 1100 | **53** |
| CHEMBL2321965 | O=C1Nc2c(Br)cccc2/C1=C1/Nc2ccc(/C=N\O)cc2/C1=N\O | 600 | **54** |
| CHEMBL2321964 | O=C1Nc2c(Br)cccc2/C1=C1/Nc2ccc(CO)cc2/C1=N\O | 400 | **55** |
| CHEMBL2321951 | COC(=O)c1ccc2c(c1)C(=N\O)/C(=C1/C(=O)Nc3ccccc31)N2 | 400 | **56** |
| CHEMBL2321948 | O=C1Nc2c(Br)cccc2/C1=C1/Nc2ccc(/C=N/O)cc2C1=O | 200 | **57** |
| CHEMBL2321961 | COC(=O)c1ccc2c(c1)NC(=C1\C(=O)Nc3c1cccc3C(F)(F)F)/C2=N/O | 100 | **58** |

*(Continued)*

**Table 1.** (Continued)

| Chembl_id | Canonical_smiles | Standard value (1C50nM) | Structure No. |
|---|---|---|---|
| CHEMBL2321952 | O=C1Nc2ccccc2/C1=C1/Nc2ccc(C(=O)O)cc2/C1=N\O | 70 | **59** |
| CHEMBL409450 | O=C1Nc2cc(Br)ccc2/C1=C1/Nc2ccccc2/C1=N\O | 5 | **60** |
| CHEMBL4128587 | NC(=O)c1c(O)cc(O)c2c1oc1c(=O)[nH]cnc12 | 10000 | **61** |
| CHEMBL4129279 | NC(=O)c1c(O)cc(O)c2c1oc1c(=O)[nH]c(=O)[nH]c12 | 10000 | **62** |
| CHEMBL4127885 | NC(=O)c1c(O)cc(O)c2c1oc1c(N)ncnc12 | 10000 | **63** |
| CHEMBL475816 | CC(=O)C1=C(O)C=C2Oc3c(C(N)=O)c(O)cc(O)c3[C@]2(C)C1=O | 10000 | **64** |
| CHEMBL4167852 | CCOC(=O)c1c2[nH]c3ccccc3c2c(-c2ccc(NC(=O)CCN(C)C)cc2)c2c(=O)[nH][s+]([O-])c12 | 2800 | **65** |

## Molecular docking investigation

The initial step in molecular docking was to obtain the SMILES strings for 12 structures reported in Fig 2 using ChemDraw software. Additionally, the 3D structure of pfGSK3β (Fig 5A) was downloaded from the Protein Data Bank, which includes a co-crystallised ligand (referred to as S1 in this study). The position of the co-crystallised ligand within the receptor aids in identifying the binding site (Fig 5B). To understand the shape of the binding pocket, a surface mapping study was conducted using MOE software. This surface mapping revealed the shape of the binding pocket (Fig 5C).

Molecular docking is a computational method that enables us to model interactions between ligands and proteins based on the lock and key theory (Fig 6). Using this method, we can predict the binding ability of molecules (ligands) to their specific proteins (receptors) and analyse how the ligand might fit within the conformational space [33], with the main goal being to identify the most stable receptor-ligand complex through their binding affinities.

## Ligand-receptor interactions

The following ranking system was used to determine which Ligand would be best suited for the receptor.

1. Molecules with many interactions, including both hydrophilic and hydrophobic ones, and very high binding energy will be rated the best.

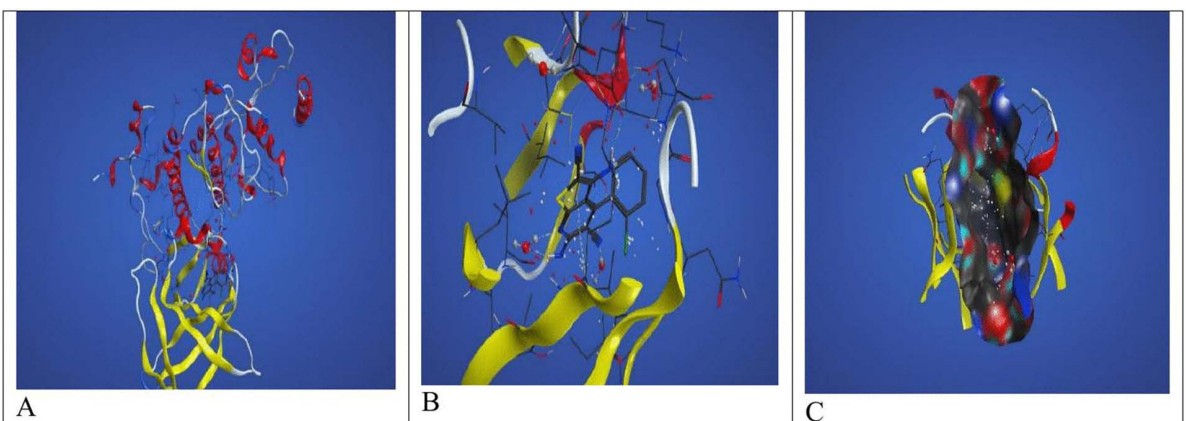

**Fig 5. Interaction of the Co-crystallised Ligand in the binding pocket of 3ZDI.** 4A relates to the structure of 3ZDI embedding the co-crystallised ligand; 4B shows the ligand position in the binding pocket, and 4C shows the surface mapping of ligand and the receptor binding pocket.

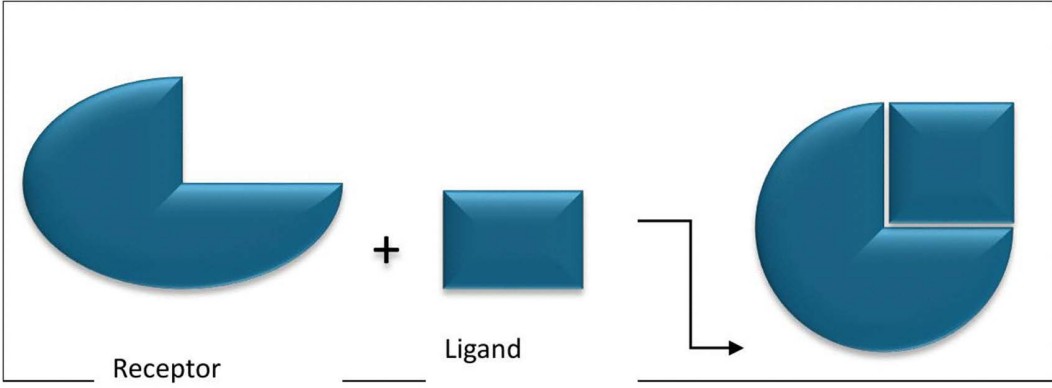

**Fig 6. Lock and key theory [34].**

Hydrophobic interactions, such as pi-pi bonds, provide thermodynamic stability and increase the tendency to form contacts among themselves rather than with polar water [35]. Meanwhile, hydrophilic interactions, such as hydrogen bond acceptors and donors, can enhance binding specificity and affinity between the ligand and the protein binding pocket [36].

2. Low binding energy despite numerous interactions, including both electrophilic and hydrophobic ones, was rated second.

3. Molecules without interactions were not taken into account.

Using the above ranking criteria, the top four molecules identified by molecular docking analysis are listed in Table 2. For simplicity of discussion, the following keywords will be employed. The IUPAC names were verified using the PubChem search (https://pubchem.ncbi.nlm.nih.gov/).

S20 –CHEMBL ID 1910196 (4-[5-(6-hydroxy-1H-indol-2-yl)pyridin-3-yl]benzonitrile).

S39- CHEMBL ID 2321945- (2-(7-bromo-2-hydroxy-1H-indol-3-yl)-3-oxoindole-6-carboxylic acid).

S56-CHEMBL ID 2321951 (methyl 2-(2-hydroxy-1H-indol-3-yl)-3-nitroso-1H-indole-5-carboxylate).

S1- (3,6-diamino-4-(2-chlorophenyl)thieno[2,3-b]pyridine-2,5-dicarbonitrile)

The molecular docking results showed that, among the 12 molecules reported in Fig 2, only the co-crystallised ligand S1 ranks among the top four molecules suitable for further investigation as a treatment for malaria. It is worth emphasising that all 12 molecules served as a standard for developing a theory that some of the molecules mined from the CHEMBL database can be repurposed to treat malaria. The molecules labelled S56, S39, and S20 were obtained from mining the CHEMBL database using Python code that we developed.

Table 2 clearly shows that the co-crystallised ligand has a higher binding affinity (s = −7.1157074) than S56 (s = −5.64057302), S39 (s = −5.42528439), and S20 (s = −5.26445723). However, this does not prove its clear superiority over the other three molecules. A closer look at Table 2 indicates that molecule S56 forms the most ligand interactions within the binding pocket (twelve interactions), while S1 interacts with ten, and S39 and S20 each with eight. Furthermore, based on the results in Table 2, we cannot conclude that S1 performs better than S39 and S20 because, when considering the amino acids involved in interactions within the binding pocket, it is evident that S1 interacts with the fewest amino acids, specifically ASP200 and VAL70. S56 performs better, with six different amino acid interactions including ILE 562, ASN 186, GLN 185, ARG 141, VAL 70, and THR 138. Of the remaining two ligands, S39 behaves similarly to S1 by interacting with only two amino acids: VAL 70 and GLN 185, while S20 interacts slightly better than S1 by engaging three amino acids: ASN 64, VAL 70, and GLN 185. Therefore, based on the molecular docking results, it can be concluded that S56 might be successfully repurposed for the inhibition of pfGSK3β, as it competes favourably in silico with the already

**Table 2. Protein Ligand Interactions of the four best-ranked molecules obtained during the molecular docking investigation, the Ligand Interaction diagrams are shown in Fig 15).**

| Structure (S) | No of interaction | s-value (binding affinity) | Types of Interaction |
|---|---|---|---|
| 1 | 10 | −7.1157074 | Ligand Receptor Interaction<br>N 6 OD2 ASP 200 (A) H-donor<br>N 6 OD2 ASP 200 (A) H-donor<br>N 11 O HOH 2029 (A) H-acceptor<br>N 11 O HOH 2029 (A) H-acceptor<br>N 18 O HOH 2031 (A) H-acceptor<br>N 18 O HOH 2031 (A) H-acceptor<br>5-ring CG1 VAL 70 (A) pi-H<br>6-ring CG1 VAL 70 (A) pi-H<br>5-ring CG1 VAL 70 (A) pi-H<br>6-ring CG1 VAL 70 (A) pi-H |
| 56 | 12 | −5.64057302 | Ligand Receptor Interaction<br>O 19 O ILE 62 (A) H-donor<br>O 19 O ILE 62 (A) H-donor<br>N 25 OD1 ASN 186 (A) H-donor<br>N 25 OD1 ASN 186 (A) H-donor<br>N 37 O GLN 185 (A) H-donor<br>N 37 O GLN 185 (A) H-donor<br>O 7 NH1 ARG 141 (A) H-acceptor<br>O 7 NH1 ARG 141 (A) H-acceptor<br>6-ring CG2 VAL 70 (A) pi-H<br>5-ring CG2 THR 138 (A) pi-H<br>6-ring CG2 VAL 70 (A) pi-H<br>5-ring CG2 THR 138 (A) pi-H |
| 39 | 8 | −5.42528439 | Ligand Receptor Interaction<br>O 24 O HOH 2029 (A) H-acceptor<br>O 24 O HOH 2029 (A) H-acceptor<br>6-ring CG2 VAL 70 (A) pi-H<br>5-ring CB GLN 185 (A) pi-H<br>6-ring CG GLN 185 (A) pi-H<br>6-ring CG2 VAL 70 (A) pi-H<br>5-ring CB GLN 185 (A) pi-H<br>6-ring CG GLN 185 (A) pi-H |
| 20 | 8 | −5.26445723 | Ligand Receptor Interaction<br>N 30 O ASN 64 (A) H-donor<br>N 30 O ASN 64 (A) H-donor<br>N 1 O HOH 2029 (A) H-acceptor<br>N 1 O HOH 2029 (A) H-acceptor<br>6-ring CG1 VAL 70 (A) pi-H<br>6-ring CB GLN 185 (A) pi-H<br>6-ring CG1 VAL 70 (A) pi-H<br>6-ring CB GLN 185 (A) pi-H |

tested S1 molecule. Nonetheless, molecular docking data alone should not be regarded as definitive bioinformatics evidence for repurposing a potential drug molecule; hence, molecular dynamics simulations of these four molecules were performed to understand how their movement within the binding pocket influences receptor stability.

## Molecular dynamics simulations

Molecular docking results alone cannot predict the success of in silico testing because they do not account for protein flexibility during protein-ligand interactions [37,38]. In this research, we focus on an in silico approach to discovering potential anti-malarial drugs that could be developed into new drug moieties for malaria treatment. The docking scores have identified a potential drug that performs better than the bioassayed drug molecules (S1). To gain a deeper understanding of

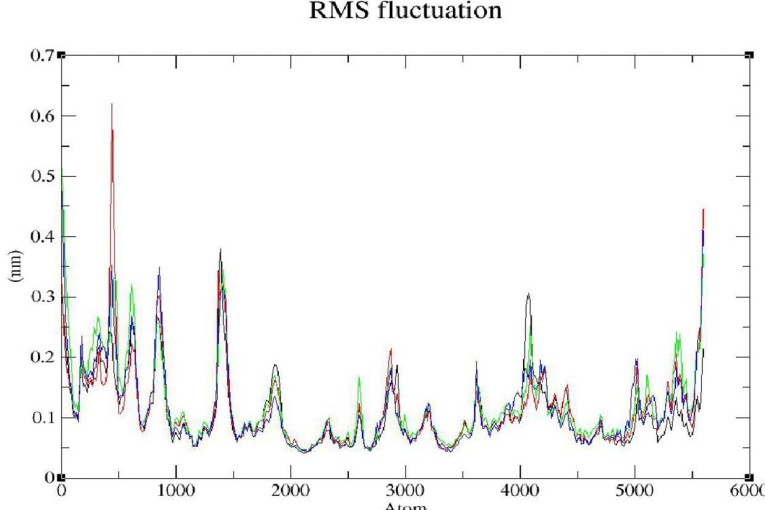

**Fig 7. RMSF results of the ligands within the receptor during the simulation.** The black graph represents the S20 ligands, the red graph depicts the S1 ligands, the green graph shows the S39 ligands, and the blue graph indicates the S56 ligand fluctuations within the receptor.

the binding behaviour of the biomolecules, molecular dynamics simulations were conducted. These simulations were run on GROMACS 2023.1 using the CHARMM all-atom force field, for a duration of 100 ns (50000000 steps). The Root Mean Square Fluctuation (RMSF, Fig 7) and Root Mean Square Deviation (RMSD, Fig 8) trajectory analyses of the MD simulations are colour-coded: S20 = black, S1 = red, S39 = green, S56 = blue. Finally, snapshots of the molecular motions were captured at 10 ns, 40 ns, 70 ns, 80 ns, 90 ns, and at the end of the simulation run (100 ns) (Figs 9–14).

Root Mean Square fluctuation highlights a region in the protein-ligand interaction where the ligand caused a significant change in the protein structure. The instability of the co-crystallised ligand between 0–1000 atoms further supported the

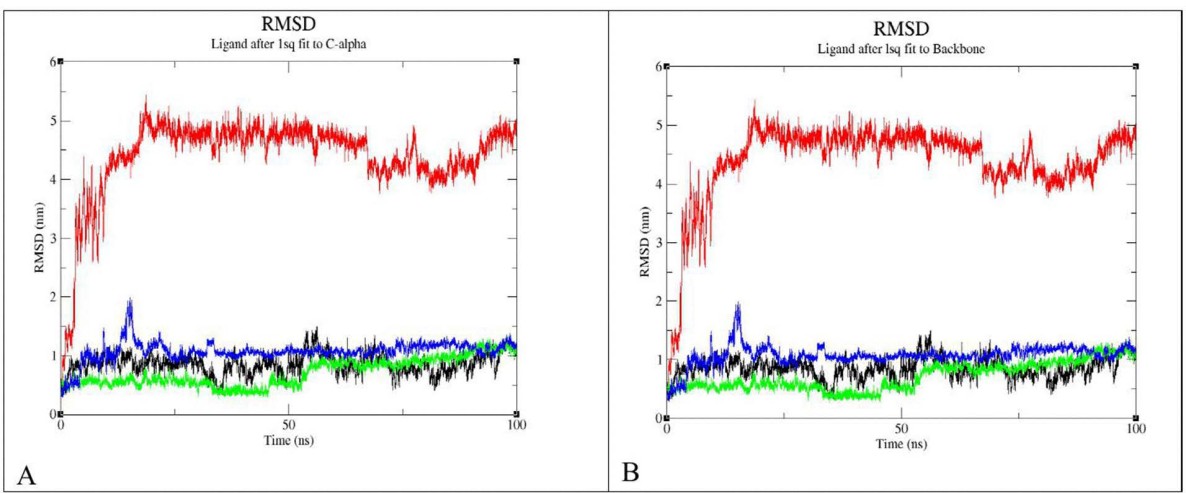

**Fig 8. RMSD results of the ligands during the molecular dynamics simulation study.** (A) Shows the RMSD analysis of the ligands in fit to the C-alpha, and (B).Shows the RMSD analysis of the ligands fit to the backbone. The black graph represents the S20 ligands, the red graph depicts the S1 ligands, the green graph shows the S39 ligands, and the blue graph indicates the S56 ligand root mean square deviations relative to the C-alpha and the backbone of the receptor.

**Snapshot at 10ns**

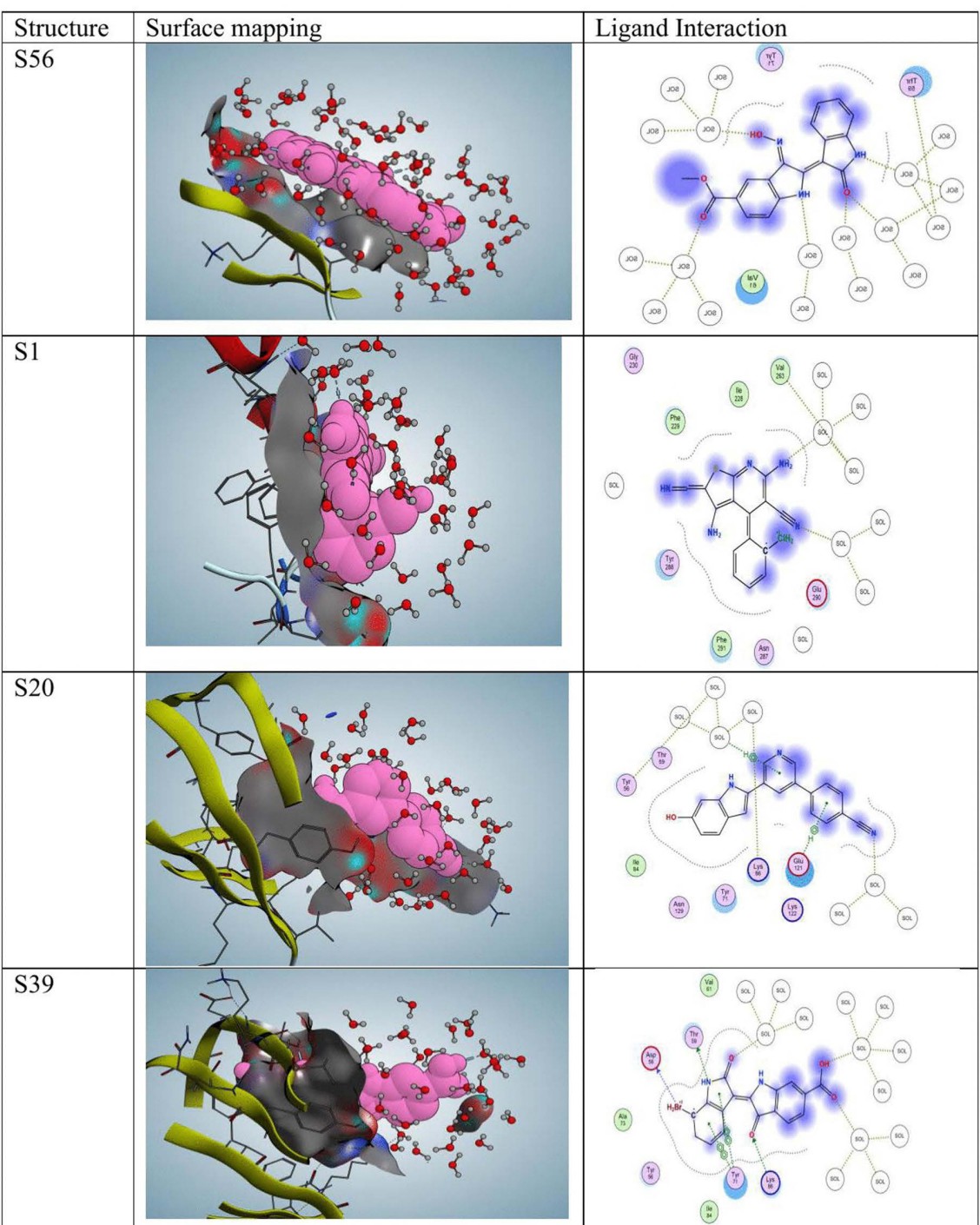

**Fig 9. Snapshot of the Molecular Dynamics simulations at 10 ns.**

**S40 Simulation results**

| Structure | Surface mapping | Ligand Interaction |
|-----------|-----------------|--------------------|
| S56 | | |
| S1 | | |
| S20 | | |
| S39 | | |

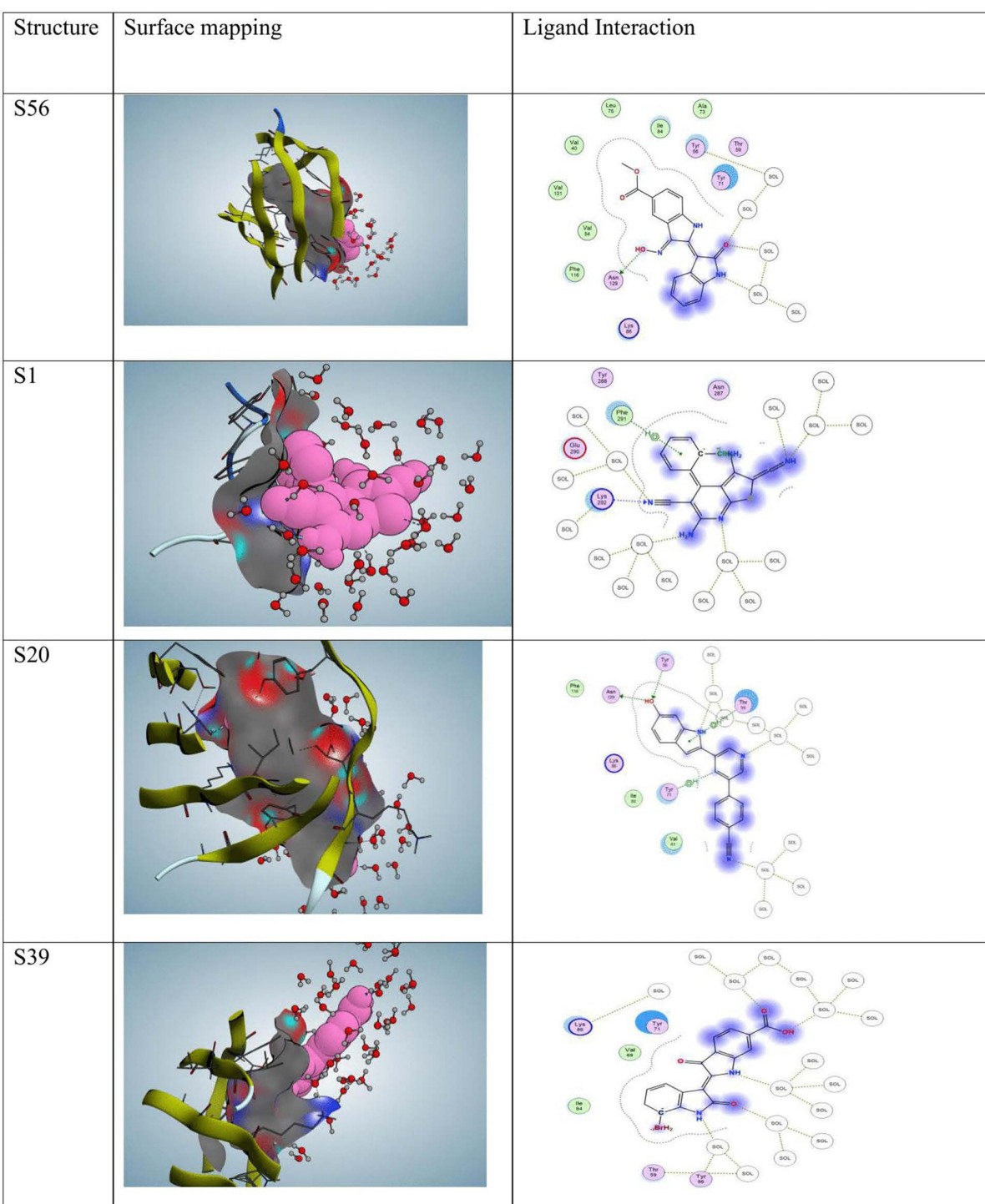

**Fig 10. Snapshot of the Molecular Dynamics Simulations at 40 ns.**

**Simulation at 70 ns**

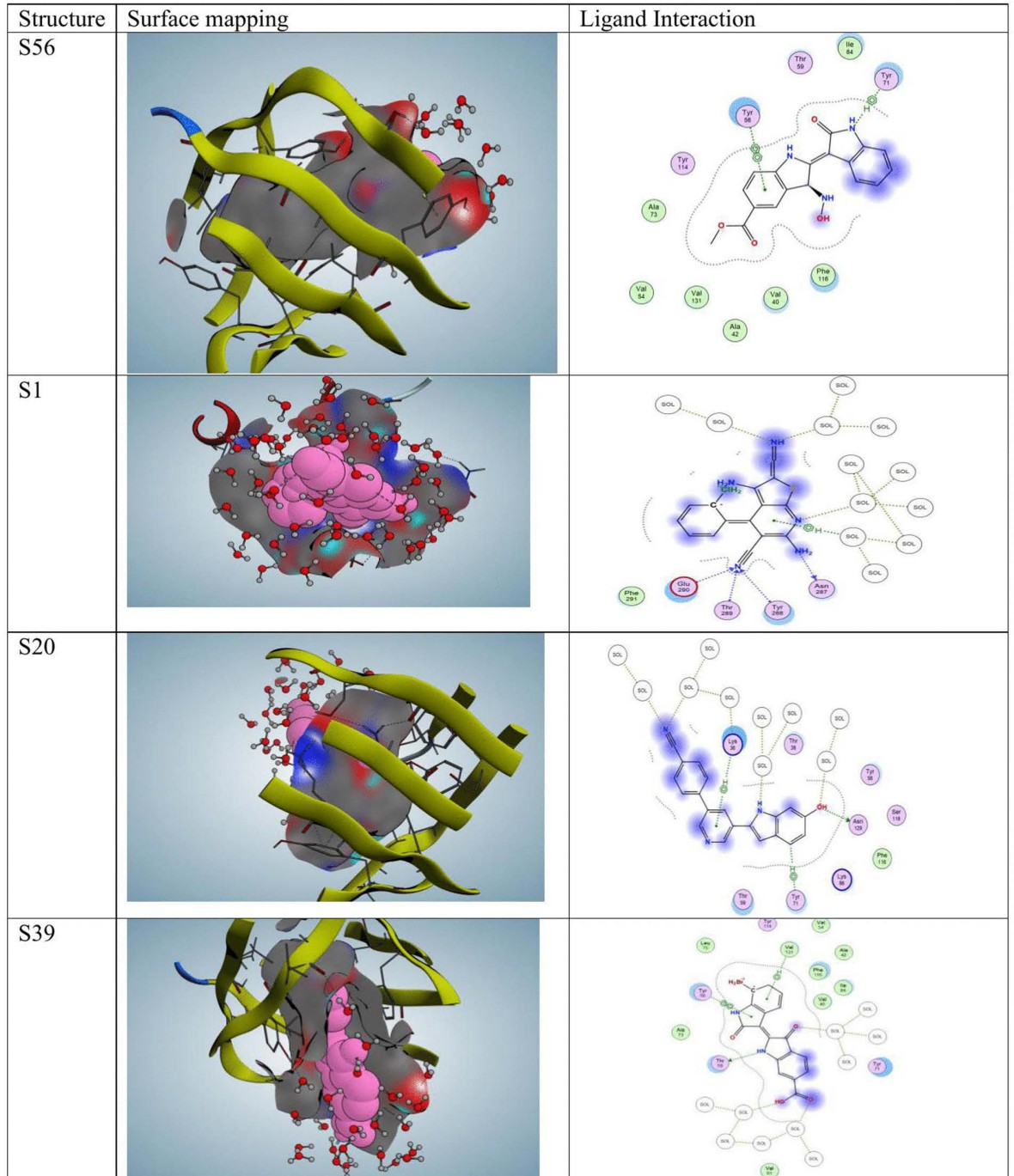

| Structure | Surface mapping | Ligand Interaction |
|---|---|---|
| S56 | | |
| S1 | | |
| S20 | | |
| S39 | | |

**Fig 11. Snapshot of the Molecular Dynamics Simulations at 70 ns.**

**Simulation at 80 ns**

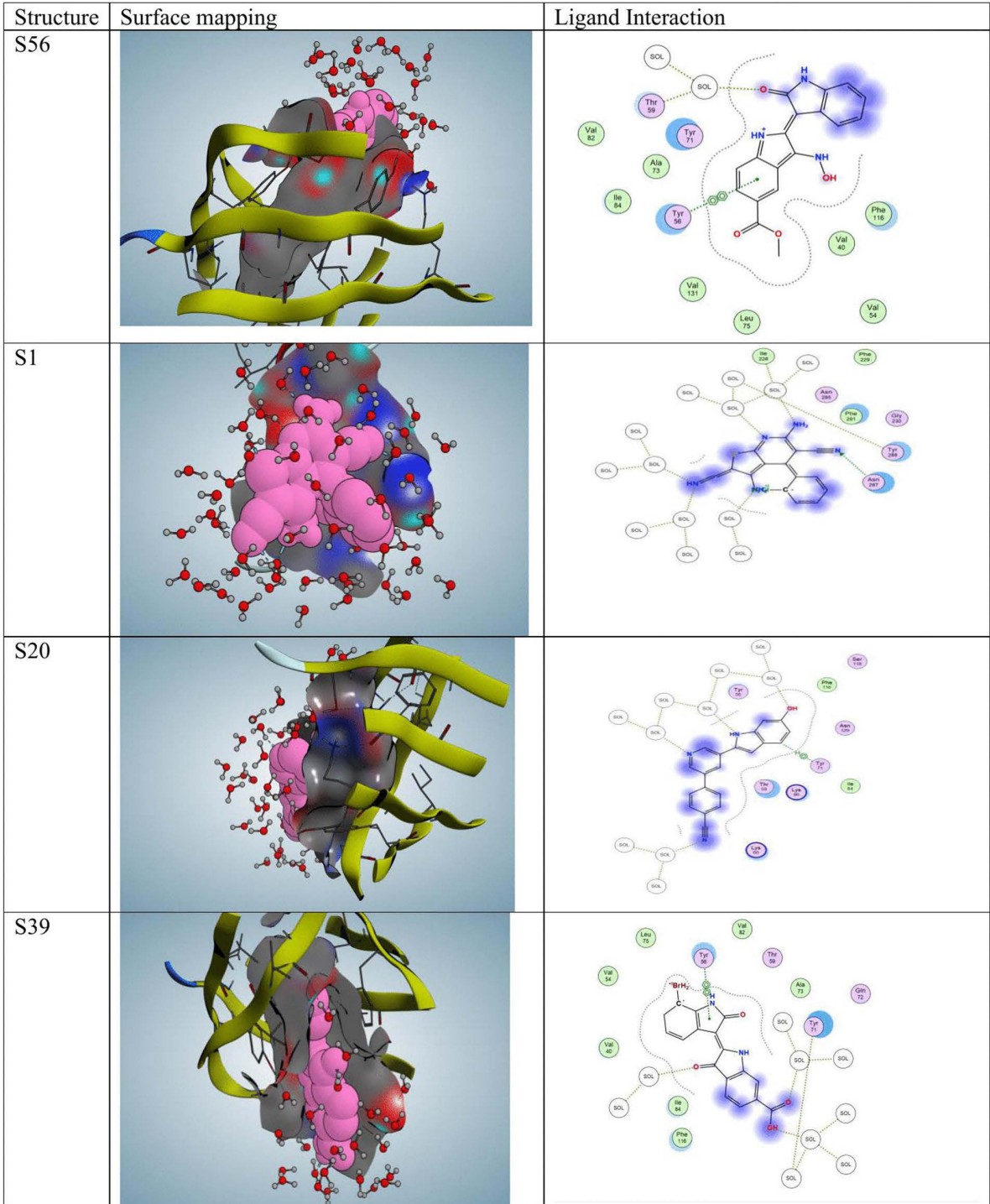

| Structure | Surface mapping | Ligand Interaction |
|---|---|---|
| S56 | | |
| S1 | | |
| S20 | | |
| S39 | | |

**Fig 12. Snapshot of the molecular dynamics simulations at 80 ns.**

**Simulation at 90 ns**

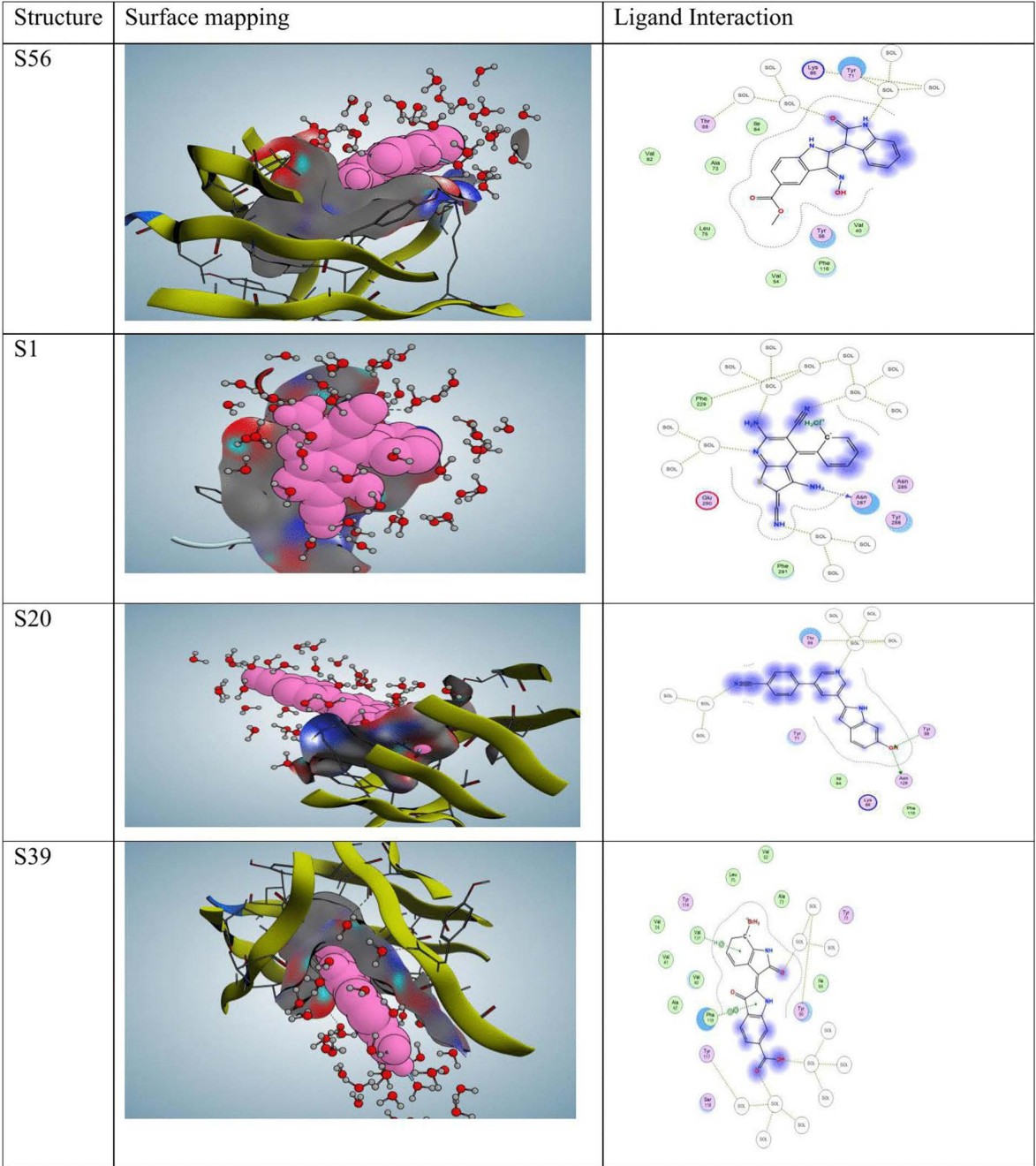

| Structure | Surface mapping | Ligand Interaction |
|-----------|-----------------|--------------------|
| S56 | | |
| S1 | | |
| S20 | | |
| S39 | | |

**Fig 13. Snapshot of the molecular dynamics simulations at 90 ns.**

**Simulation at 100 ns**

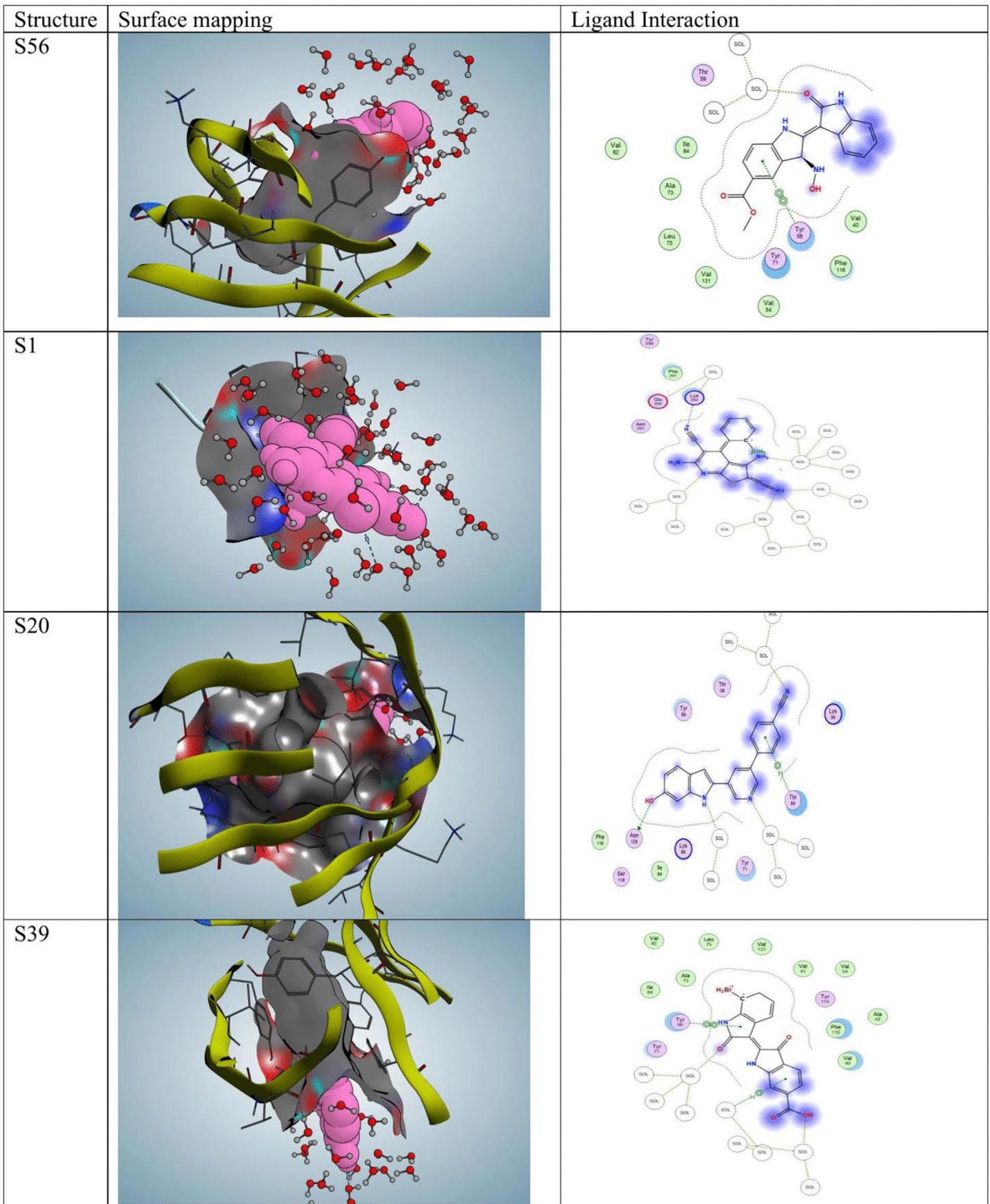

**Fig 14. Snapshot of the Molecular Dynamics Simulations at 100 ns.**

behaviour observed in the RMSD plot (Fig 8). The repurposed drug molecules exhibited relatively low fluctuation within the binding pocket.

The conformational stability of the protein-ligand complex was indicated by the root mean square deviation (RMSD) of the complex. Fig 8 shows that the trajectory of the co-crystallised ligand (S1) has a very high RMSD when interacting with the backbone or the C-alpha chain. Meanwhile, the repurposed compounds showed very low RMSD values between 0 and 2 nm, with deviations nearing convergence at 1.0 nm. The RMSD value is important because the lower it is, the more compact the protein-ligand interaction, indicating a better lock-and-key fit. This improved interaction also supports the molecular docking results, which showed better ligand interactions for compounds S56, S20, and S39. Therefore, these molecules might exhibit better inhibitory properties than S1 against pfGSK3β.

Snapshots of simulations were captured at various intervals – 10 ns, 40 ns, 70 ns, 80 ns, 90 ns, and 100 ns. These snapshots help us understand the ligand's behaviour within the binding pocket throughout the simulation. According to Marco De Vico et al. [39], snapshots validate docking results; molecular dynamics simulations indicate a poor molecular docking study because they are likely to produce a trajectory in which the ligand appears to leave the binding pocket. In contrast, a good molecular docking result will show the ligand remaining within the binding pocket during the Molecular Dynamics Simulations production run. This behaviour can also be visualised further using the RMSF and RMSD diagrams shown in Figs 7 and 8 above.

At 10 ns simulations step (Fig 9) S56 and the co-crystallised ligand S1 did not interact with any amino acids in the binding pocket, but with the water in the system. S20 undergoes hydrophobic interactions (pi-H) with GLU 121, while the rest interact with solvent in the environment. S39 ligand exhibits better interactions with the receptors than all other ligands at 10 ns. It forms hydrogen bonds (H-donor) with THR 59 and ASP 58, and an additional hydrogen bond (H-acceptor) with LYS 86. Moreover, S39 ligand shows a hydrophobic interaction (pi-pi) between the 5-ring of the ligand and the 6-ring of TYR 71, as well as a pi-pi interaction between the 6-membered ring of the ligand and the 6-membered ring of TYR 71. Notably, although the co-crystallised ligands S1 and S56 did not interact with any amino acids in the binding pocket at 10 ns, this did not render them less effective. Furthermore, Marco Devivo and co-workers [39], observed that water molecules can influence ligand binding to the receptor, either facilitating or preventing it. Fortunately, at all snapshots at the 10 ns time step, the ligands were not rejected from the pfgsk3β binding pocket.

At 40 ns Simulations step (Fig 10), S56 interacted with ASN 129, while S1 formed hydrophilic interactions (hydrogen acceptor) with LYS 292. Meanwhile, the six-membered ring of ligand S1 engaged in hydrophobic interactions (π-H) with PHE 291. S20 formed hydrophobic interactions with the six-membered ring of TYR 71 (H-π), and the five-membered ring of ligand S20 interacted with THR 59 (π-H). Furthermore, S20's interaction with the receptor was characterised by hydrogen bond donor interactions with ASN 129, while the remaining interactions occurred with the surrounding solvent. S39 does not exhibit any interactions with amino acids at this stage, only with the solvents present in the environment.

At 70 ns Molecular Dynamics Simulations step (Fig 11), S56 still undergoes hydrophobic interactions through its N2 18 with the 6-ring TYR 71 (A) H-pi, and there is another hydrophobic interaction between the ligand's 6-membered ring and the six-membered ring of TYR 56 (pi-pi). S1 interacts through hydrophilic interactions (H-donor) with ASN 287 and acts as a hydrogen bond acceptor with TYR 288 and 289, as well as GLU 290. S20 forms hydrogen bonds (H-donor) with ASN 129 and engages in hydrophobic interactions involving the ligand's 6-ring with TYR 71 (H-pi) and pi-H interactions involving the ligand's 6-ring and LYS 36. The remaining interactions are with the solvent environment. S39 forms a hydrogen bond (H-donor) with THR 59, a hydrophobic (pi-H) interaction with VAL 131, and another hydrophobic (pi-pi) interaction between the ligand's 5-membered ring and the 6-membered ring of TYR 56.

The snapshot of the interaction at 80 ns simulations step (Fig 12) showed that the 6-membered ring of S56 ligand undergoes hydrophobic (pi-pi) interactions with the 6-membered ring of TYR 51 in the receptor. S1 only has a hydrogen bond acceptor with the amino acid ASN 287, while the rest are with the solvent in the environment. The S20 ligand only undergoes hydrophobic (pi-H) interactions with the 6-membered ring of TYR 71, while the remaining interactions are with

the solvents in the environment. S39 ligands only undergo hydrophobic (pi-pi) interactions between the 5-membered ring of the ligand and the 6-membered ring of TYR 56.

Furthermore, at 90 ns Molecular Dynamics Simulations step (Fig 13) it was observed that S56 did not interact with the receptor but with water in the environment. At this simulation point, S1 acted as a hydrogen-bond donor to ASN 287, while the others interacted with water. S20 formed a hydrogen bond (H-donor) with ASN 129 and an H-acceptor with TYR56, as well as with solvents in the environment. S39 engaged in a hydrophobic (pi-H) interaction between the ligand's 6-membered ring and VAL 131, and a hydrophobic (pi-pi) interaction involving the ligand's 5-membered ring and the 6-membered ring of PHE 116 of the receptor.

Finally, the snapshot of the protein-ligand interaction at 100 ns (Fig 14) revealed that Ligand S56 still maintains a hydrophobic interaction (pi-pi) involving the six-membered ring of the ligand and TYR56. S1 forms a hydrogen bond with LYS292, while the others interact with the solvent in the environment. S20 ligands engage in hydrogen bonding (H-donor) with ASN129 and hydrophobic interaction (pi-H) with THR59; the remaining interactions are with solvent. The only amino acid S39 interacted with at the end of the simulation was a hydrophobic (pi-pi) interaction between the five-membered ring of the ligand and the six-membered ring of TYR56.

**g_mmpbsa calculation results**

The limitations of molecular docking analysis can also be addressed further by supporting the Molecular Dynamics Simulations with the MMPBSA calculation. MMPBSA calculations decompose protein-ligand interactions into the sum of all non-bonded interactions. The contributions of Van der Waals energy (ΔVDWAALS), Gas Phase Electrostatic energy (ΔG GAS), solvation energy (ΔG Solv), and total energy (ΔG Total) were estimated for the S1-, S20-, S39-, and S56-protein complexes using MM-PBSA, as shown in Table 3. The binding free energies for these complexes are all negative. The results clearly show that these compounds exhibit significant binding affinity for pfGSK3β; in particular, S56 still demonstrates its superiority over the co-crystallised ligand, with a higher binding affinity (−14.45 kJ mol$^{-1}$).

**In silico toxicity profile**

In the era of ML (Machine Learning) and the application of Artificial Intelligence to facilitate research, it is noteworthy that thousands of laboratory mice are sacrificed for testing the toxicity profile of potential drug molecules, yet less than 1% of these molecules reach the market. The use of in silico toxicity studies could serve as an important tool for examining the biological toxicity of new drug molecules and might reduce the costs associated with purchasing laboratory rats for toxicological testing. The in silico toxicity study was conducted using a web server tool, Protox III. The reliability and accuracy of this tool have been widely described [32],

Table 4 gave the overall toxicity predictions and log P of the drug molecules The overall toxicity prediction indicated that all compounds could be moderately toxic, except for compounds S6 and S20, which showed a predicted toxicity of 3. However, the toxicity spectrum shown in Fig 3, 15, 16 suggests that no single biological molecule is entirely eliminated.

Table 3. g_mmpbsa analysis of the protein-ligand complex of the S1-, S20, S39-, and S56- protein-ligand complex.

| Structure | Average Vander Waals (KJ mol$^{-1}$) ΔVDWAALS | Average Electrostatic Energy (KJ mol$^{-1}$) ΔGGAS | Solvation Energy (KJ mol$^{-1}$) ΔG Solv | Total Energy ((KJ mol$^{-1}$)) ΔGTotal |
|---|---|---|---|---|
| S1 | −20.88 | −32.74 | 19.71 | −13.03 |
| S20 | −30.45 | −45.74 | 20.61 | −12.45 |
| S39 | −29.35 | −30.75 | 19.98 | −13.80 |
| S56 | −32.46 | −32.45 | 19.67 | −14.45 |

**Table 4. In silico Toxicity Prediction, PA represent the Prediction accuracy, and AS represents the average similarity, toxicity profile ( 1 2 3 4 5 6 ).**

| Structure | Predicted Toxicity class | logPO/W |
|---|---|---|
| 1 | 4 (Average Similarity=42.69% and Prediction Accuracy 54.26% | 4.69 |
| 2 | 4 (Average Similarity=55.62% and Prediction Accuracy=67.38%) | 2.93 |
| 3 | 4 (AS = 36.67%) and PA=23% | 0.86 |
| 4 | 4 (AS = 49.46% and PA=54.26%) | 4.23 |
| 5 | 4 (AS = 52.06% and PA=67.38% | 3.9 |
| 6 | 3(AS = 38.39%) and PA=23% | 4.94 |
| 7 | 4 (AS = 41.09%) and PA=54.26% | 4.72 |
| 8 | 4 (AS = 37.74%) and PA=23% | 6.65 |
| 9 | 4 (AS = 39.58%) and PA=23% | 6.7 |
| 10 | 4(AS = 39.61%) and PA=23% | 7.35 |
| 11 | 4 (AS = 38.39%) and PA=23% | 7.06 |
| 12 | 4(AS = 38.05%) and PA=23% | 6.81 |
| 56 | 4 (AS = 49.05%) and PA=54.26%) | 2.72 |
| 39 | 4 (AS = 50.64%) and PA=67.38% | 3.39 |
| 20 | 3(AS = 57.08% and PA=67.38% | 4.47 |

Instead, the spectrum aims to present what might be expected when assessing the pharmacokinetics of the potential drug molecules in vivo and in vitro.

## Conclusion

This research utilised an in silico approach to repurpose molecules that inhibit GSK3β, a protein found in humans, for targeting the pfGSK3 enzymes in malaria parasites. The ligand interactions of molecules S56, S39, and S20 demonstrated strong interactions, comparable to S1, and better than all other eleven compounds shown in Fig 2. Furthermore, the molecular dynamics simulation results indicated that these three promising pfGSK3β inhibitors perform better than the tested molecule (S1) in silico, based on RMSF and RMSD analyses. As a result, these drugs warrant further investigation for the treatment of malaria infections.

The in silico toxicity studies of these molecules showed that the compounds exhibited their respective toxicity spectra, which will serve as a useful guide in designing in vivo and in vitro pharmacological studies. According to Masch and co-worker [9], pfgsk- 3β (pdb.3 zdi) is a viable target in the development of novel drugs for treating malaria infection; therefore, we have successfully compiled Python codes that enabled us to mine the CHEMBL database for GSK- 3β inhibitors developed for other diseases. This data mining yielded 53 molecules, of which we identified three potential candidates. Molecular docking, molecular dynamics simulations, and molecular mechanics Poisson- Boltzmann surface area (MMPBSA) data analysis identified three promising drug molecules: S 20- CHEMBLID 1910196 (4-[5-(6- hydroxy- 1 H- indol- 2- yl) pyridin- 3- yl]benzonitrile), S 39- CHEMBL ID 2321945 (2-(7- bromo- 2- hydroxy- 1 H- indol- 3- yl)- 3- oxindole- 6- carboxylic acid), and S 56- CHEMBL ID 2321951 (methyl 2-(2- hydroxy- 1 H- indol- 3- yl)- 3- nitroso- 1 H- indole- 5- carboxylate),which could inhibit pfgsk 3 β. Compound S 56 showed better in silico performance than S 1 – (3, 3, 3,6- diamino- 4-(2- chlorophenyl)thieno[2, 3- b] pyridine- 2, 5- dicarbonitrile), the co- crystallised ligand in pfgsk 3 β used as a control. The binding affinities for S1 and S56 are – 7. 1157074 (10 ligand interactions) and – 5. 64057302 (12 ligand interactions), respectively. The MD simulations yielded average root- mean- square deviations (RMSDs) of 4. 5 nm for S 1 and 1. 0 nm for S 56. Additionally, the root mean square fluctuation (RMSF) of S1 showed greater fluctuation between 0–1000 atoms compared to S56. MMPBSA analysis revealed comparable total energies: S56 at- 14. 45 kJ/mol and S 1

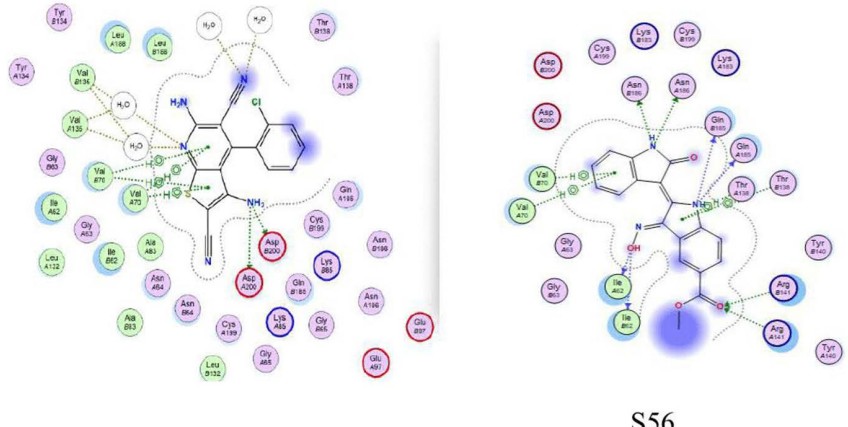

S1

S56

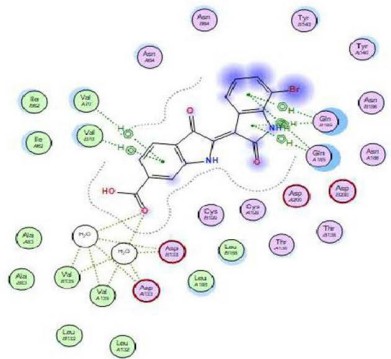

S39

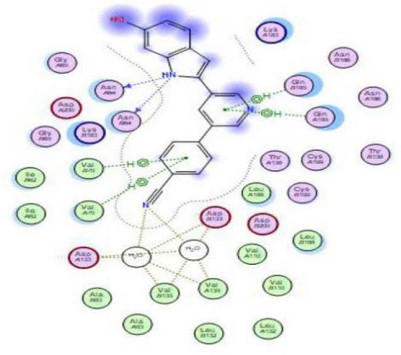

S20

**Fig 15. Ligand Interactions diagram of the four best molecules obtained during the molecular docking, the interactions details are shown in table 2.**

| Structure | Organ Toxicity | | | | | | | | | | | Metabolism | | | | | |
|---|---|---|---|---|---|---|---|---|---|---|---|---|---|---|---|---|---|
| | Hepatotoxicity | Neurotoxicity | Nephrotoxicity | Respiratory toxicity | Cardiotoxicity | Carcinogenicity | Immunotoxicity | Mutagenicity | Cytotoxicity | BBB-barrier | Ecotoxicity | Cytochrome CYP1A2 | Cytochrome CYP2C19 | Cytochrome CYP2C9 | Cytochrome CYP2D6 | Cytochrome CYP3A4 | Cytochrome CYP2E1 |
| 1 | Active (63%) | Active (60%) | Inactive (78%) | Inactive (51%) | Inactive (93%) | Active (50%) | Inactive (98%) | active (63%) | Inactive (63%) | Active (62%) | Inactive (50%) | Active (66%) | Active (52%) | Inactive (54%) | Active (54%) | Active (67%) | Inactive (97%) |
| 2 | Active (54%) | Active (53%) | Active (50%) | Active (52%) | Inactive (75%) | Active (64%) | Inactive (83%) | Active (53%) | Inactive (75%) | Active (78%) | Inactive (55%) | Active (73%) | Active (63%) | Active (64%) | Inactive (56%) | Active (71%) | Inactive (99%) |
| 3 | Inactive (72%) | Active (68%) | Inactive (60%) | Active (78%) | Inactive (76%) | Inactive (57%) | Inactive (93%) | Inactive (57%) | Inactive (66%) | Active (75%) | Inactive (64%) | Inactive (76%) | Inactive (78%) | Inactive (60%) | Inactive (55%) | Inactive (85%) | Inactive (98%) |
| 4 | Inactive (58%) | Active (88%) | Inactive (71%) | Active (66%) | Inactive (78%) | Inactive (55%) | Inactive (69%) | Inactive (69%) | Inactive (73%) | Active (88%) | Active (58%) | Active (93%) | Inactive (68%) | Active (53%) | Active (78%) | Inactive (84%) | Inactive (99%) |
| 5 | Inactive (55%) | Active (50%) | Inactive (65%) | Active (60%) | Inactive (72%) | Active (68%) | Inactive (57%) | Active (78%) | Inactive (57%) | Active (84%) | Inactive (50%) | Inactive (64%) | Inactive (70%) | Inactive (59%) | Inactive (70%) | Inactive (65%) | Inactive (99%) |
| 6 | Inactive (55%) | Active (63%) | Inactive (84%) | Active (73%) | Inactive (88%) | Inactive (58%) | Inactive (99%) | Active (78%) | Inactive (59%) | Active (73%) | Active (60%) | Active (57%) | Inactive (64%) | Active (54%) | Active (60%) | Inactive (57%) | Inactive (94%) |
| 7 | Active (64%) | Active (66%) | Inactive (61%) | Inactive (50%) | Inactive (87%) | Active (53%) | Inactive (99%) | Inactive (51%) | Active (60%) | Active (73%) | Active (54%) | Inactive (68%) | Inactive (74%) | Inactive (55%) | Inactive (54%) | Inactive (53%) | Inactive (99%) |
| 8 | Active (66%) | Active (59%) | Inactive (64%) | Inactive (51%) | Inactive (89%) | Active (53%) | Inactive (99%) | Active (56%) | Inactive (51%) | Active (71%) | Active (53%) | Active (56%) | Inactive (62%) | Inactive (53%) | Active (56%) | Active (61%) | Inactive (99%) |
| 9 | Active (67%) | Active (60%) | Inactive (65%) | Inactive (52%) | Inactive (88%) | Active (53%) | Inactive (97%) | Active (56%) | Inactive (53%) | Active (71%) | Active (53%) | Active (57%) | Inactive (62%) | Inactive (52%) | Active (56%) | Active (62%) | Inactive (98%) |
| 10 | Active (67%) | Active (60%) | Inactive (65%) | Inactive (52%) | Inactive (88%) | Active (53%) | Inactive (98%) | Active (66%) | Inactive (53%) | Active (71%) | Active (53%) | Active (57%) | Inactive (62%) | Inactive (52%) | Active (56%) | Active (62%) | Inactive (98%) |
| 11 | Active (73%) | Active (63%) | Inactive (65%) | Active (53%) | Inactive (89%) | Active (51%) | Inactive (96%) | Active (51%) | Inactive (62%) | Active (73%) | Active (59%) | Active (50%) | Inactive (60%) | Inactive (51%) | Active (52%) | Active (56%) | Inactive (99%) |
| 12 | Active (67%) | Active (60%) | Inactive (65%) | Inactive (51%) | Inactive (88%) | Active (52%) | Inactive (93%) | Active (57%) | Inactive (51%) | Active (70%) | Active (53%) | Active (56%) | Inactive (62%) | Inactive (51%) | Active (56%) | Active (62%) | Inactive (98%) |
| 56 | Active (52%) | Inactive (77%) | Active (64%) | Active (53%) | Inactive (72%) | Inactive (52%) | Inactive (58%) | Active (51%) | Inactive (57%) | Inactive (57%) | Inactive (64%) | Inactive (62%) | Inactive (70%) | Active (62%) | Inactive (72%) | Inactive (55%) | Inactive (99%) |
| 39 | Active (59%) | Inactive (53%) | Active (75%) | Inactive (51%) | Inactive (83%) | Active (54%) | Inactive (66%) | Inactive (57%) | Inactive (61%) | Inactive (60%) | Inactive (67%) | Inactive (70%) | Inactive (85%) | Active (65%) | Inactive (73%) | Inactive (71%) | Inactive (99%) |
| 20 | Active (52%) | Inactive (58%) | Inactive (58%) | Inactive (59%) | Inactive (84%) | Inactive (64%) | Inactive (94%) | Active (55%) | Inactive (59%) | Active (78%) | Active (56%) | Active (51%) | Inactive (61%) | Active (58%) | Inactive (58%) | Inactive (68%) | Inactive (97%) |

**Fig 16. Toxicity profile of the molecules shown in Figure 2 and the top three molecules identified through data mining of the CHEMBL database.** The green profile indicates that the compounds are inactive in the toxicity category, while the red profile shows they are active.

at- 13. 03 kJ/mol. An in silico toxicity study using Protox III indicated the possible toxicity of the repurposed compounds. In conclusion, we propose that molecules S 39, S 20, and S 56 could be repurposed as potential anti- malarial drugs. Therefore, (S 56, S 39, and S 20) could be repurposed for inhibiting pfgsk- 3β enzymes found in the malaria parasite. S56 is the most superior among these, which is why its structure is shown in the graphical abstract.

## Strengths and limitations

The strength of this study lies in utilising an in silico screening approach, which enabled rapid and cost-effective evaluation of novel glycogen synthase inhibitors. This computer-based method facilitated the potential identification of drug targets for malaria treatment. Protein-ligand molecular docking is a vital tool for understanding how ligands interact with receptors, indicating which amino acids are targeted to halt disease progression. This structure-based drug design effectively ranks protein-ligand interactions by score and ligand interaction type [40,41], assisting researchers in selecting hit molecules for further in silico, in vivo, and in vitro testing. Additionally, molecular dynamics simulations have become an essential investigative technique to verify experimental data outside of in silico models [42–46], making them an important in silico method for repositioning biological molecules.

Most challenges stem from the fact that an in silico environment might not fully replicate biological systems, as true as this may be. However, advances in Machine Learning and Artificial Intelligence have made in silico studies a vital theoretical approach for predicting the activity of potential molecules, especially in drug repurposing. Furthermore, the main challenges identified include the high computational demand of in silico testing, such as Molecular Dynamics Simulations, and the approximations inherent to the force fields used in these calculations. The development of high-performance

computing systems, along with powerful desktop and laptop computers, has addressed the issue of high computational demand. This research was conducted on a high-end yet affordable computer (£4000) with an Intel Core i9 processor (24 cores) and 32 GB of RAM, capable of running approximately 300 ns/day according to the log file. The second major limitation, related to the accuracy of force fields, has led researchers to incorporate Quantum Mechanical calculations into force field development [47]. While the protox III toxicity can be limited by intermodal variability and dependence on structural variability, it is found to be helpful in providing rapid, cost-effective alternatives for pre-wet laboratory toxicity screening of chemical substances [48].

## Future work

This study successfully achieved its aim, and we conducted an in silico investigation to identify novel compounds S56, S20, and S39, which, based on the computational analysis in this research, show potential for repositioning as pfGSK3β inhibitors for malaria treatment. However, further work is required to advance and validate these findings, including in vitro and in vivo analyses of these molecules, before their progression through clinical trial phases.

## Supporting information

**S1 File. Appendix.**
(DOCX)

## Author contributions

**Conceptualization:** Kassim Folorunsho Adebambo.

**Data curation:** Kassim Folorunsho Adebambo.

**Investigation:** Sara Otify.

**Methodology:** Kassim Folorunsho Adebambo, Sara Otify.

**Writing – original draft:** Kassim Folorunsho Adebambo, Sara Otify.

**Writing – review & editing:** Kassim Folorunsho Adebambo.

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
