## [Decision Letter · Decision Letter 0]

1 Sep 2025

Dear Dr. Adebambo,

Thank you for submitting your manuscript to PLOS ONE. After careful consideration, we feel that it has merit but does not fully meet PLOS ONE’s publication criteria as it currently stands. Therefore, we invite you to submit a revised version of the manuscript that addresses the points raised during the review process.

The reviewers commented major modifications on the paper, hence, the authors are requested to do the corrections. 

We look forward to receiving your revised manuscript.

Kind regards,

Sajjad Ahmad

Academic Editor

PLOS ONE

Journal Requirements:

Additional Editor Comments (if provided):

Reviewer #1: The reviewers commented major modifications on the paper, hence, the authors are requested to do the corrections.

Reviewer #2: The reviewers commented major modifications on the paper, hence, the authors are requested to do the corrections.

Reviewers' comments:

Reviewer's Responses to Questions

**Comments to the Author**

1. Is the manuscript technically sound, and do the data support the conclusions?

Reviewer #1: Yes

Reviewer #2: Yes

2. Has the statistical analysis been performed appropriately and rigorously?

Reviewer #1: Yes

Reviewer #2: Yes

3. Have the authors made all data underlying the findings in their manuscript fully available?

Reviewer #1: Yes

Reviewer #2: Yes

4. Is the manuscript presented in an intelligible fashion and written in standard English?

Reviewer #1: No

Reviewer #2: Yes

Reviewer #1: 1. Reassess ligand ranking using binding free energy (MM-PBSA/MM-GBSA) to support conclusions.

2. Revise claims in the conclusion regarding compound superiority.

3. Conduct professional language editing to correct grammar and improve clarity.

4. Replace all instances of "insilico" with correctly spaced "in silico".

5. Move technical installation/code instructions to supplementary material or methods appendix.

6. Reorganize and clarify figures/tables with proper legends and concise captions.

7. Summarize MD snapshots using key RMSD/RMSF graphs rather than detailed timepoint text.

8. Add a clear limitation section acknowledging toxicity and lack of wet-lab validation.

9. Rewrite the abstract to be more concise, highlighting objectives, methods, key results, and conclusions in <300 words.

Reviewer #2: Reviewer comments

The authors present a very interesting manuscript titled “Insilico investigation of novel Plasmodium Falciparum Glycogen Synthase Kinase(pfGSk3β) inhibitors for the treatment of malaria infection”

” The presented study results are very interesting.

1 The manuscript is written clearly, and sections are well planned by the authors.

2 There are some minor changes regarding some typos and grammatical errors. Make sure that the citation is properly cited and add some updated citations.

3 Abstract should be rewritten with incorporation of proper computational results.

4 Introduction section is very short please make sure it has updated citations and some current investigation regarding the same topic portion must be revisited.

5 Figure 3 and Figure 5 are not in a proper readable format, make sure it is easy to understand.

6 Incorporate the pdb id in the methodology section as well with proper citation of the published articles.

7 Proper scientific format for species names is required throughout the journal.

8 Add legends of the compounds in the RMSD and RMSF graph for better understanding.

**Do you want your identity to be public for this peer review?** For information about this choice, including consent withdrawal, please see our Privacy Policy

Reviewer #1: No

Reviewer #2: No

---

## [Author Response · Author response to Decision Letter 1]

30 Oct 2025

I have corrected the journal based on the valuable comments provided by the reviewers, there was no major methodological or discussion issue. All the comments were about organisations of the manuscripts, grammar and re-writing of the Abstract. This has been done and the corrections included in the revised manuscript with tracking.

---

## [Decision Letter · Decision Letter 1]

17 Nov 2025

Insilico investigation of novel Plasmodium Falciparum Glycogen Synthase Kinase(pfGSk3β) inhibitors for the treatment of malaria infection

PONE-D-25-30659R1

Dear Dr. Adebambo,

We’re pleased to inform you that your manuscript has been judged scientifically suitable for publication and will be formally accepted for publication once it meets all outstanding technical requirements.

Kind regards,

Sajjad Ahmad

Academic Editor

PLOS ONE

Additional Editor Comments (optional):

The reviewers comments are well adjusted in the revised manuscript, hence, the paper can be accepted for publication.

Reviewers' comments:

Reviewer's Responses to Questions

**Comments to the Author**

Reviewer #1: All comments have been addressed

Reviewer #2: All comments have been addressed

2. Is the manuscript technically sound, and do the data support the conclusions?

Reviewer #1: Yes

Reviewer #2: Yes

3. Has the statistical analysis been performed appropriately and rigorously?

Reviewer #1: Yes

Reviewer #2: Yes

4. Have the authors made all data underlying the findings in their manuscript fully available?

Reviewer #1: Yes

Reviewer #2: Yes

5. Is the manuscript presented in an intelligible fashion and written in standard English?

Reviewer #1: Yes

Reviewer #2: Yes

Reviewer #1: All necessary changes has been made. Proceed for acceptance if other reviewers suggestions are incorporated.

Reviewer #2: (No Response)

**Do you want your identity to be public for this peer review?** For information about this choice, including consent withdrawal, please see our Privacy Policy

Reviewer #1: No

Reviewer #2: No

---

## [Editor Report · Acceptance letter]

PONE-D-25-30659R1

PLOS ONE

Dear Dr. Adebambo,

I'm pleased to inform you that your manuscript has been deemed suitable for publication in PLOS ONE. Congratulations! Your manuscript is now being handed over to our production team.

Kind regards,

on behalf of

Dr. Sajjad Ahmad

Academic Editor

PLOS ONE